# St. John’s Wort Suppresses Growth in Triple-Negative Breast Cancer Cell Line MDA-MB-231 by Inducing Prodeath Autophagy and Apoptosis

**DOI:** 10.3390/nu12103175

**Published:** 2020-10-17

**Authors:** Mikyoung You, Young-Hyun Lee, Hwa-Jin Kim, Ji Hyun Kook, Hyeon-A Kim

**Affiliations:** 1Department of Nutrition, University of Massachusetts Amherst, Amherst, MA 01003, USA; budi1030@naver.com; 2Department of Food and Nutrition, Mokpo National University, Jeollanam-do 58554, Korea; ciesl7@naver.com (Y.-H.L.); tailove2212@naver.com (H.-J.K.); kookji76@daum.net (J.H.K.)

**Keywords:** triple-negative breast cancer, autophagy, apoptosis, St. John’s Wort

## Abstract

The rational regulation of programmed cell death by means of autophagy and apoptosis has been considered a potential treatment strategy for cancer. We demonstrated the inhibitory effect of St. John’s Wort (SJW) on growth in the triple-negative breast cancer (TNBC) cell line and xenografted mice and its target mechanism concerning autophagic and apoptotic cell death. SJW ethanol extract (SJWE) inhibited proliferation in a dose-dependent manner. SJWE treatment dramatically increased autophagy flux and apoptosis compared with the control. The autophagy inhibitor, 3-methyladenine (3-MA), reversed the SJWE-induced inhibition of cell proliferation and regulation of autophagy and apoptosis, indicating that SJWE induced apoptosis through prodeath autophagy. Furthermore, SJWE inhibited tumor growth and induced autophagy and apoptosis in the tumor of MDA-MB-231 xenografted athymic nude mice. Our results indicate that SJWE might have great potential as a new anticancer therapy for triple-negative breast cancer by inducing prodeath autophagy and apoptosis.

## 1. Introduction

Two different types of programmed cell death, autophagy and apoptosis, are crucial to chemotherapy in cancer treatment. Autophagy, which is a catabolic process of self-digestion, plays a key role in various physiopathological processes, such as cell death and survival, development, and tumorigenesis [1,2,3]. There are two faces of autophagy in cancer pathogenesis: tumor suppression and cellular survival. Recent studies have indicated that autophagy activation induced by anticancer treatment results in cancer cell death and autophagy defection increases cancer cell growth [4,5]. In contrast, some studies showed that autophagy-deficient cancer cells exhibit decreased survival in tumors [6,7]. Meanwhile, apoptosis is a crucial cytotoxic mechanism of anticancer agents. Disturbances caused by defects in apoptosis mechanisms might eventually cause cells to extend their lifespan, contributing to neoplastic expansion independently of cell division [8]. Since growth inhibition is the primary target of chemotherapy and irradiation, failure to regulate the inducing of prodeath autophagy and apoptotic pathways can make cancer cells resistant to therapy, posing additional challenges. 

Breast cancers that are negative to estrogen receptor α (ERα), progesterone receptor (PR), and hormone epidermal growth factor receptor 2 (HER2/neu) are termed “triple-negatives”. Current treatment strategies have a limitation in that patients with triple-negative breast cancer (TNBC) still show poor survival because of resistance to chemotherapy [9]. Some researchers showed that TNBC exhibits a very wide spectrum of genomic profiles, mutation status of oncogenes, and clonal frequencies [10]. Because of this heterogeneity, TNBCs are clinically aggressive and associated with overall poor prognosis. Despite developments in early detection and additional therapeutic approaches, effective treatment for TNBCs is still lacking [11,12,13,14]. In this context, intensive efforts should be made to find therapeutic targets for these more aggressive forms of breast cancer. 

St. John’s Wort (SJW), *Hypericum perforatum*, is a herbaceous plant native to Europe and Asia [15]. It has been traditionally used to treat mild to severe depressive disorders [16,17,18]. A variety of bioactive compounds of SJW ethanol extract (SJWE), including hypericin and hyperforin, have anticancer effects [19,20,21,22,23,24]. In addition, we previously reported that SJWE regulated proliferation and apoptosis in MCF-7 breast cancer cells by inhibiting the AMPK/mTOR and Bcl-2 family pathways [25]. However, the anticancer effect of SJWE in terms of autophagy and apoptosis in TNBC cells is not understood yet. In this study, we hypothesized that SJWE suppresses cancer progression in the TNBC cell line MDA-MB-231, in vitro and in vivo, by inducing prodeath autophagy and apoptosis.

## 2. Materials and Methods

### 2.1. Reagents

SJWE was prepared using a previously established protocol [25,26] and stored at −20 °C. The triple-negative breast cancer (TNBC) cell line of MDA-MB-231 was purchased from the American Type Culture and Collection (ATCC, Rockville, MD, USA). Dulbecco’s modified Eagle’s medium (DMEM) and fetal bovine serum (FBS) were purchased from Gibco (BRL, Gaithersburg, MD, USA). The Muse^TM^ Annexin V and Dead Cell Kit was supplied by Merck KGaA (Darmstadt, Germany). Primary antibodies were obtained from Cell Signaling Technology (Beverly, MA, USA) and Santa Cruz Biotechnology Inc. (Dallas, TX, USA). Horseradish peroxidase-conjugated secondary antibodies and a Western blot enhanced chemiluminescence kit were obtained from Jackson ImmunoResearch Inc. (West Grove, PA, USA) and IMGENEX (San Diego, CA, USA), respectively. 

### 2.2. Cell Culture, Cell Viability, and Proliferation

MDA-MB-231 cells were maintained in DMEM supplemented with penicillin 100 U/mL, streptomycin 100 μg/mL, and 10% FBS. To check cell viability, cells were treated with SJWE for 24 h and then living cells were stained using 3-(4,5-dimethulthiazol-2-yl)-2,5-diphenyltetrazolium bromide (MTT). For proliferation assay, cells were seeded at a density of 6 × 10^4^ cells/well and the medium was replaced with or without SJWE for 5 days. Cells were harvested and counted. The results are presented as a percentage of the control of each group.

### 2.3. Apoptosis Assay

To assess apoptosis, MDA-MB-231 cells were incubated with either dimethylsulfoxide(DMSO) (0.1% of media) or SJWE for 24 h. Cells were washed with PBS, collected, and centrifuged at 450 g for 5 min and resuspended with complete medium. Muse^TM^ Annexin V and Dead Cell reagent (100 μL) was added to resuspended cells, which were incubated at RT for 20 min. The resuspended cells were assayed using the Muse^TM^ Cell Analyzer. 

### 2.4. Immunoblotting

Western blotting was performed as described previously using primary antibodies (LC3, p62, Atg3, Atg5, Atg5-12, Beclin1, PI3Kp85, p-PI3Kp85(Tyr458)/p55(Tyr199), Akt, p-Akt, Bcl-2, Bcl-xL, Bad, p-Bad, Bax, mTOR, p-mTOR, AMPK, p-AMPK, and β-actin) [25]. Protein expression was detected using a chemiluminescent substrate and quantified with the UVP imaging system (UVP, Upland, CA, USA) using AlphaEaseFC 4.0 (Alpha Innotech, San Leandro, CA, USA). The details of antibody information are available in Appendix A.

### 2.5. Xenograft Mouse Study

The study protocol was approved by the Institutional Animal Care and Use Committee of Mokpo National University (MNU-IACUC-2015-35-12). Athymic nude mice (Balb/c) were purchased from OrientBio. (Seongnam-si, Kyunggi-do, Korea) and housed in high-efficiency and particulate air-filter cages in a pathogen-free facility with a 12 h light/dark cycle. Mice were fed an AIN-93G diet (7% corn oil replacing soybean oil, from Research Diets, New Brunswick, NJ, USA) for the entire experimental period. After acclimation, mice were subcutaneously (s.c.) inoculated in the bilateral flank with Matrigel (BD Biosciences, San Jose, CA, USA) containing 2 × 10^6^ MDA-MB-231 cells. Palpable and measurable tumors were found beginning 7–8 days after cells were injected. At 3 weeks after inoculation, tumors had grown at 90% of the injection site. Mice were randomly grouped to receive s.c. injection (*N* = 10 in each group) of 50 mg/kg of either SJWE or PBS (C) for 4 weeks (five times a week). We monitored food intake, body weight, and palpable tumor diameters weekly. We calculated tumor volumes as (π/6) × [length (mm) × width^2^ (mm^2^)]. Seven weeks after cell inoculation, mice were sacrificed by cervical dislocation and tumors were dissected for further examination. 

### 2.6. H&E Staining and TUNEL Assay

An isolated tumor mass was divided into small pieces, fixed in 10% neutral buffered formalin, and embedded in paraffin. Tissues were then cut into slices with a thickness of 4 µm. Slices on slides were then stained with hematoxylin and eosin (H&E staining) followed by microscopy. Apoptosis in tumor tissues was detected using a TUNEL assay kit (Promega, Madison, WI, USA) according to the manufacturer’s instructions and analyzed them under a microscope (original magnification: ×200). The TUNEL-positive cells were quantified as relative folds.

### 2.7. Statistical Analysis

Statistical analyses were performed using SPSS version 23.0 (SPSS Inc., Chicago, IL, USA). In vitro results are expressed as mean ± S.E. from three independent experiments. Comparisons were based on one-way analysis of variance (ANOVA) followed by Duncan’s multiple range test. In the animal study, Student’s *t*-test was used to identify the significant difference between control and SJWE-treated groups. *p* < 0.05 was considered statistically significant.

## 3. Results

### 3.1. SJWE Inhibited the Proliferation in TNBC MDA-MB-231 Cells

MDA-MB-231 cells were cultured in the presence or absence of SJWE and examined for cell viability and proliferation. SJWE showed no apparent toxicity to MDA-MB-231 cells treated initially for 24 h (Figure 1A). However, treatment with SJWE for five days resulted in a dose-dependent decrease in the proliferation of MDA-MB-231 cells (*p* < 0.05) (Figure 1B). 

### 3.2. SJWE Induced Pro-Death Autophagy in TNBC Line MDA-MB-231 Cells

During the process of autophagy, LC3-I is transformed into LC3-II and then involved in the formation of autophagosomes [27]. We assessed the modulation of autophagy by SJWE in TNBC line MDA-MB-231 cells. LC3-II was dose-dependently upregulated, and the LC3-II/LC3-I ratio was gradually increased by SJWE treatment. Another key protein, p62, also decreased in the SJWE-treated cells. (Figure 2A). In addition, autophagy-related 5 (Atg5) was increased, whereas Atg12 was reduced by SJWE treatment. Nonetheless, SJWE showed an increase in the expression of the Atg 5–12 complex (Figure 2B) and Beclin-1. These results indicate that SJWE increased autophagosome formation in TNBC line MDA-MB-231 cells. 

### 3.3. SJWE Induces Apoptosis via the Downregulation of Antiapoptic Proteins in TNBC MDA-MB-231 Cells

To investigate the effect of SJWE on apoptosis, we measured the apoptotic cell population in TNBC line MDA-MB-231 cells. The apoptotic cells, lower-right and upper-right quadrants, were dose-dependently induced by SJWE treatment (*p* < 0.05) (Figure 3A). There was a greater reduction in antiapoptotic proteins in the SJWE groups than in the control, including B-cell lymphoma 2 (Bcl-2), B-cell lymphoma-extra large (Bcl-xL), and p-Bcl-2-associated death promoter (p-bad) (Figure 3B). However, SJWE did not affect the expression level of the proapoptotic protein Bad, whereas it decreased the expression of Bax (Figure 3B). Furthermore, the p-PI3K/PI3K and p-mTOR/mTOR ratios were significantly suppressed by SJWE treatment (Figure 3C). Taken together, these results indicate that SJWE induces apoptosis by inhibiting antiapoptotic proteins. 

### 3.4. The Inhibition of Autophagy Reversed Apoptosis Induced by SJWE in TNBC Line MDA-MB-231 Cells

To investigate whether autophagy is involved in SJWE-induced apoptosis, MDA-MB-231 cells were treated with 50 ug/mL SJWE in the presence or absence of the autophagy inhibitor, 3-methyladenine (3-MA, 1 mM). As expected, 3-MA treatment combined with SJWE antagonized the SJWE-induced autophagy activation, which was assessed by LC3-II/LC3-I and p62 expression in the TNBC line MDA-MB-231. We also found that 3-MA with SJWE significantly decreased Beclin-1 expression, which was upregulated by SJWE treatment (Figure 4A). In addition, the inhibition of autophagy dramatically increased the expression of antiapoptotic proteins Bcl-2 and Bcl-xL (Figure 4B). To test whether the inhibition of autophagy attenuates the SJWE-induced suppression of cell proliferation, MDA-MB-231 cells were treated with SJWE and 3-MA for three or five days. The combined treatment with 3-MA and SJWE for both three and five days significantly increased cell proliferation compared with SJWE treatment alone (Figure 4C and Appendix A). 

Emerging evidence shows that PI3K/mTOR signaling is related to increasing autophagy and apoptosis and is always activated in cancer including breast cancer [28,29]. Based on our observation that SJWE was involved in the activation of PI3K (Figure 3C), we asked whether SJWE was involved in regulating autophagy via the downregulation of the PI3K/mTOR pathway. As expected, the treatment of 3-MA combined with SJWE markedly increased the p-PI3K/PI3K ratio to the level of the control. Furthermore, p-mTOR protein expression was partially increased by 3-MA treatment in MDA-MB-321 cells (Figure 4D). Taken together, these results indicate that SJWE increases apoptotic cell death by inducing autophagy.

### 3.5. SJWE Inhibited Tumor Progression by Inducing Autophagic and Apoptotic Cell Death in MDA-MB-231 Cells Xenografted Athymic Nude Mice 

Next, we explored whether the SJWE attenuates tumor growth in mice inoculated with TNBC line MDA-MB-231 cells. Mice were transplanted with MDA-MB-231 cells in Matrigel in the bilateral flan. Three weeks after inoculation, mice were injected with SJWE for four weeks. During SJWE treatment, the body weights of animals were not significantly different between the control and SJWE-treated mice (Figure 5A). At three weeks after treatment (at six weeks after inoculation), tumor volumes were significantly decreased in the SJWE-treated mice compared to those in the control mice (*p* < 0.05) (Figure 5B). Conforming to the suppressed tumor growth, the number of apoptotic cells was dramatically higher in the SJWE-treated mice (Figure 5C). Next, we detected the protein levels of autophagic and apoptotic cell death. We found that SJWE treatment markedly elevated LC3-II accumulation, but suppressed p62 expression. In addition, Beclin-1 expression was significantly increased in the SJWE-treated mice (Figure 5D). Consistent with the in vitro study, antiapoptotic proteins, Bcl-2 and Bcl-xL, were lowered by SJWE treatment (Figure 5E). Furthermore, the p-PI3K/PI3K and p-mTOR/mTOR ratio remarkably decreased in the SJWE-treated groups (Figure 5F). Collectively, these results suggest that SJWE reduces tumor growth by inducing autophagic and apoptotic cell death in MDA-MB-231 cell-inoculated mice.

## 4. Discussion

The anticancer effect of SJWE and its active compounds has been reported. These effects are thought to be mainly mediated by apoptotic cell death, especially the intrinsic pathway [19,20,21,22,23,24,25]. In this study, we demonstrated that SJWE induced programmed cell death by activating autophagy and apoptosis as evidenced by an increased autophagic flux and the growing number of apoptotic cells in the TNBC line MDA-MB-231 cells. Moreover, SJWE-induced autophagy activation and apoptosis led to the suppression of proliferation through modulation of the PI3K/mTOR pathway in the TNBC line MDA-MB-231 cells. Additionally, our in vivo study revealed a function of SJWE as an anticancer modifier in controlling tumor growth.

Autophagy is a conserved dynamic cellular activity during which cytoplasmic components including organelles or proteins are degraded and recycled to maintain basal cellular bioenergetics [30]. It has been reported that antitumor reagents and chemotherapeutics induce autophagy in several cancer models in vitro and in vivo [31,32]. We demonstrated that SJWE induced autophagy by identifying the conversion of LC3-I to LC3-II and downregulation of p62 in MDA-MB-231 cells and tumor tissues of MDA-MB-231 xenografted athymic nude mice. It was further supported by the increased expression of pro-autophagic proteins, such as Atg3 and Atg12. As another notable protein in autophagy, Beclin-1 expression was dramatically increased by SJWE treatment in vitro and in vivo. Beclin-1 plays an essential role in autophagy formation in a complex with the class III PI3K and is downregulated by Bcl-2, the antiapoptotic protein [33,34,35,36]. Our results showed that SJWE critically induced autophagy in TNBC line MDA-MB-231 cells and xenografted mice (Figure 2 and Figure 5). 

Apoptosis, type 1 programmed cell death, is regulated via intrinsic and extrinsic pathways. The Bcl-2 family, known as the intrinsic pathway, is a central player in apoptosis, which had two groups of protein including anti- and pro-apoptotic [37]. SJWE dose-dependently induced the apoptosis of MDA-MB-231 cells and increased the number of apoptotic cells in tumor tissues in MDA-MB-231 xenografted athymic nude mice. SJWE suppressed the expression levels of antiapoptotic proteins, including Bcl-2, Bcl-xL, and p-Bad, but not proapoptotic proteins. One mechanism could be the crosstalk between autophagy and apoptosis by the phosphorylation of Bcl-2 and Beclin-1 [38,39,40]. It is possible that SJWE induced Beclin-1 expression via the suppression of Bcl-2. However, it still needs further study to find out if the lack of Bcl-2 is involved in autophagy activation by SJWE treatment. 

TNBC that do not have ERα, PR, and HER2/neu are some of the most aggressive forms of breast cancer. TNBC cells cannot be treated with hormonal therapies including tamoxifen or Herceptin, a humanized antibody against HER2, because of the lack of a targeted receptor [41]. In addition, in breast cancer, there are distinct abnormalities in the apoptotic pathway such as Bcl-2 overexpression, which confers resistance to chemotherapy [42]. Nowadays, accumulating evidence indicates that apoptosis and autophagy may be simultaneously induced by antitumor compounds in order to kill tumor cells in a complex, coordinated, and cooperative manner [43,44,45]. In consideration of this, it was of interest to see how the inhibition of autophagy was involved in apoptosis by SJWE treatment. We applied autophagy inhibitor 3-MA with or without SJWE in TNBC line MDA-MB-231 cells. We found that SJWE-induced autophagy responses decreased in 3-MA treatment combined with SJWE as evidenced by the reversion of LC3-II/LC3-I and p62 expression. Combined treatment with 3-MA and SJWE significantly increased Bcl-2 and Beclin-1 expression compared with SJWE treatment alone. Moreover, 3-MA treatment combined with SJWE antagonized the SJWE-induced inhibition of cell proliferation. We currently think that the inhibition of autophagy led to the suppression of apoptosis and proliferation in MDA-MB-231 cells treated with SJWE (Figure 4).

Cumulative evidence has shown that the activation of the PI3K/mTOR signaling pathway plays an important role in the occurrence of malignant tumors; hence targeting suppression of this signaling pathway might be a promising therapeutic strategy for cancer treatment [46,47,48]. The PI3K/mTOR signaling pathway is necessary for the induction of autophagy and apoptosis [29,46]. We found that treatment with SJWE downregulated p-PI3K/PI3K and p-mTOR/mTOR ratios in MDA-MB-231 cells and in tumors of MDA-MB-231 cell-inoculated mice (Figure 3 and Figure 5). Furthermore, 3-MA treatment combined with SJWE reversed SJWE-induced dephosphorylation of PI3K and mTOR (Figure 4). Considering the inhibitory effect of 3-MA on the SJWE-induced regulation of PI3K and mTOR, and cell proliferation, SJWE-induced autophagy activation has played an important role in downregulating cell proliferation partially via the PI3K/mTOR pathway. 

One fact is that adenosine monophosphate-activated protein kinase (AMPK) plays a fundamental role in the regulation of mTOR, which increases autophagy flux [49,50]. In our study, SJWE increased AMPK activation in a dose-dependent, and SJWE with 3-MA treatment decreased p-AMPK expression, suggesting that SJWE-caused autophagy was mediated partially by regulating AMPK/mTOR signaling (Appendix A). Our study did not consider the blocking of AMPK signaling by attenuating mTOR expression, which constitutes potential caveats for future study.

## 5. Conclusions

We demonstrated that SJWE inhibited cell proliferation, induced autophagy, and apoptosis in in vitro and in vivo studies. The inhibition of autophagy attenuated apoptosis by SJWE in TNBC line MDA-MB-231 cells treated with SJWE. Moreover, our study is the first to show that SJWE causes prodeath autophagy and apoptosis by suppressing the PI3K/mTOR pathway in TNBC line MDA-MB-231 cells. Our present study shows that SJWE can be used as a promising therapeutic approach to develop an anticancer agent and to treat aggressive triple-negative breast cancer. 

## Figures and Tables

**Figure 1 nutrients-12-03175-f001:**
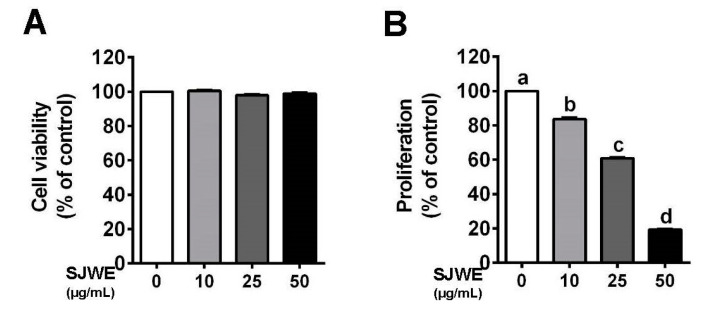
St. John’s Wort ethanol extract decreased the proliferation in triple-negative breast cancer (TNBC) line MDA-MB-231 cells. (**A**) Cell viability using 3-(4,5-dimethulthiazol-2-yl)-2,5-diphenyltetrazolium bromide (MTT) assay. (**B**) Proliferation after treatment with or without SJWE for 5 days. C: DMSO, dimethylsulfoxide (final concentration of 0.1%); SJWE 10: St. John’s Wort ethanol extract 10 μg/mL; SJWE 25: St. John’s Wort ethanol extract 25 μg/mL; SJWE 50: St. John’s Wort ethanol extract 50 μg/mL. Different letters are significantly different by Duncan’s multiple range test (*p* < 0.05).

**Figure 2 nutrients-12-03175-f002:**
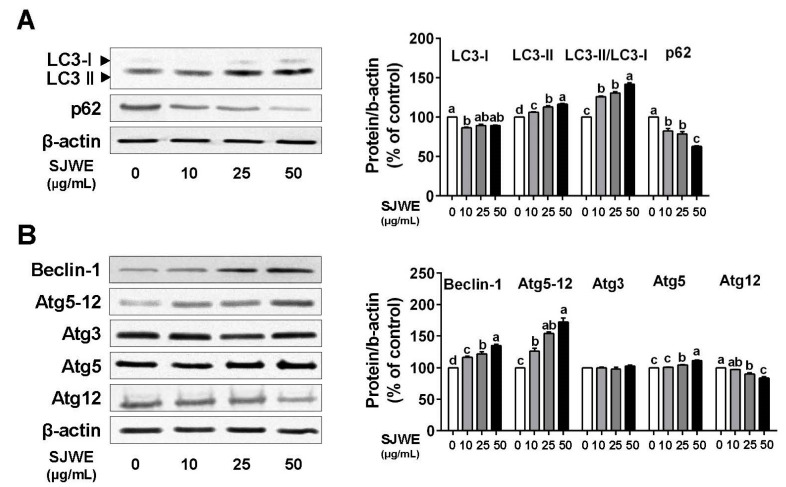
St. John’s Wort ethanol extract induces autophagy formation in TNBC line MDA-MB-231 cells. MDA-MB-231 cells were cultured with Dulbecco’s modified Eagle’s medium (DMEM) and treated with 0, 10, 25, or 50 μg/mL SJWE for 24 h. (**A**) Western blot analysis of LC3 and p62. (**B**) Western blot analysis of Beclin-1, Atg5-12, Atg3, Atg5, and Atg12. β-actin was used as a control for quantification. C: DMSO (final concentration of 0.1%); SJWE 10: St. John’s Wort ethanol extract 10 μg/mL; SJWE 25: St. John’s Wort ethanol extract 25 μg/mL; SJWE 50: St. John’s Wort ethanol extract 50 μg/mL. Different letters are significantly different by Duncan’s multiple range test (*p* < 0.05).

**Figure 3 nutrients-12-03175-f003:**
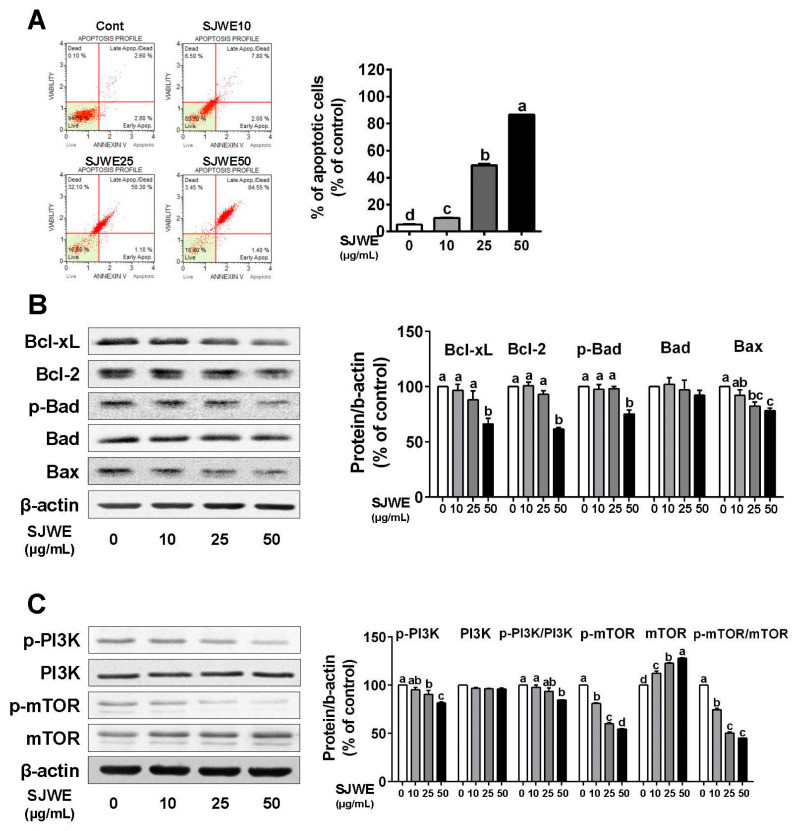
St. John’s Wort ethanol extract increases apoptosis via the downregulation of antiapoptotic proteins in TNBC line MDA-MB-231 cells. MDA-MB-231 cells were cultured with DMEM and treated with 0, 10, 25, or 50 μg/mL SJWE for 24 h. (**A**) Apoptosis using Annexin V and Dead Cell kit. Lower-right quadrant (early stages of apoptosis): Annexin V-PE (+) and Dead Cell marker (−); upper-right quadrant (late stages of apoptosis): Annexin V-PE (+) and Dead Cell marker (+). (**B**) Western blot analysis of Bcl-2 family proteins of Bcl-xL, Bcl-2, p-Bad, Bad, and Bax. (**C**) Western blot analysis of p-PI3Kp85(Tyr458)/p55(Tyr199), PI3Kp85, p-mTOR, and mTOR. β-actin was used as a control for quantification. C: DMSO (final concentration of 0.1%); SJWE 10: St. John’s Wort ethanol extract 10 μg/mL; SJWE 25: St. John’s Wort ethanol extract 25 μg/mL; SJWE 50: St. John’s Wort ethanol extract 50 μg/mL. Different letters are significantly different by Duncan’s multiple range test (*p* < 0.05).

**Figure 4 nutrients-12-03175-f004:**
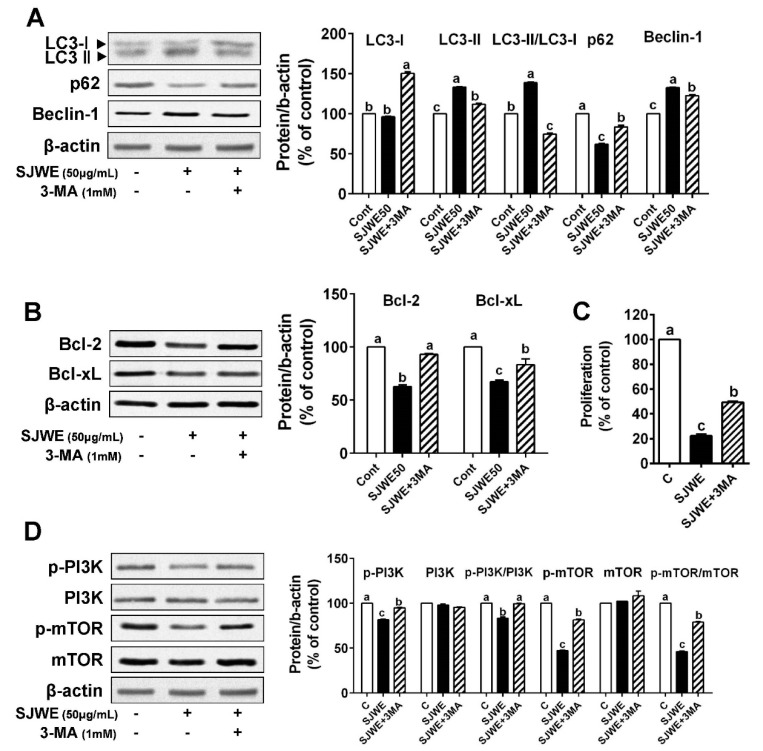
Inhibition of autophagy decreased St. John’s Wort ethanol extract-induced apoptosis in TNBC line MDA-MB-231 cells. MDA-MB-231 cells were treated with 50 μg/mL SJWE for 24 h with or without the autophagy inhibitor (3-MA, 1 mM). (**A**) Western blot analysis of the autophagy-related proteins of LC3, p62, and Beclin-1. (**B**) Western blot analysis of Bcl-2 and Bcl-XL proteins. (**C**) Proliferation with SJWE treatment in the presence and absence of 3-MA for 5 days. (**D**) Western blot analysis of p-PI3Kp85(Tyr458)/p55(Tyr199), PI3Kp85, p-mTOR, and mTOR. C: DMSO (final concentration of 0.1%); SJWE 50: St. John’s Wort ethanol extract 50 μg/mL; SJWE 50 + 3MA: St. John’s Wort ethanol extract 50 μg/mL with 3-MA treatment. Different letters are significantly different by Duncan’s multiple range test (*p* < 0.05).

**Figure 5 nutrients-12-03175-f005:**
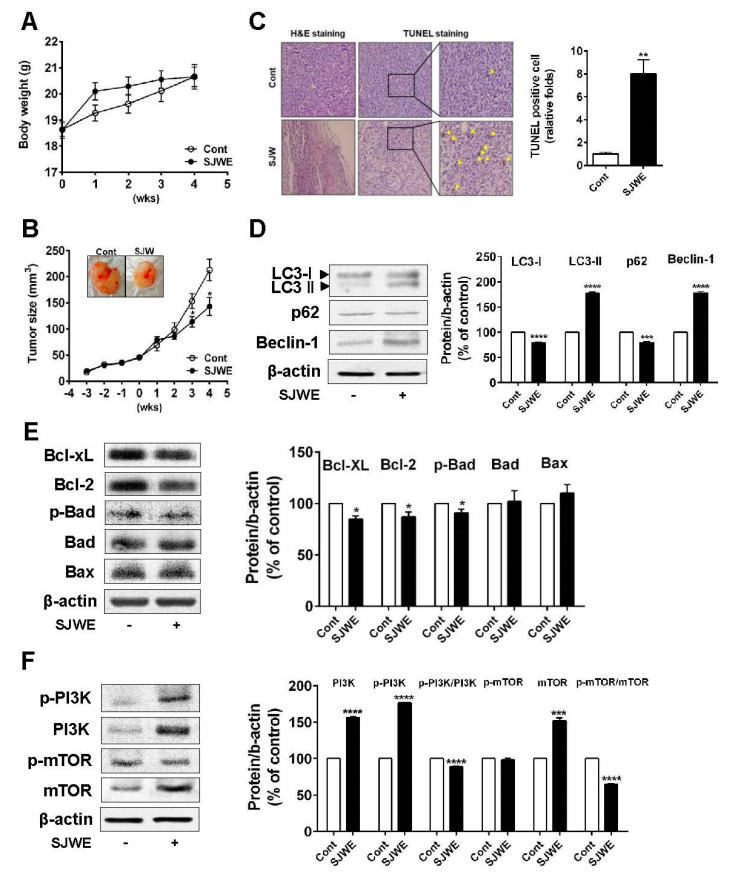
St. John’s Wort ethanol extract inhibits tumor growth through autophagic and apoptotic cell death in MDA-MB-231 cell-xenografted nude mice. Three weeks after inoculation, mice were injected with either 50 mg/kg SJWE or PBS for 4 weeks. (**A**) Body weight. (**B**) Tumor weight. (**C**) H&E staining and TUNEL assay of the tumor (arrows: TUNEL-positive cells). (**D**) Western blot analysis of autophagy-related proteins of LC3, p62, and Beclin-1. (**E**) Western blot analysis of Bcl-2 family. (**F** Western blot analysis of p-PI3Kp85(Tyr458)/p55(Tyr199), PI3Kp85, p-mTOR, and mTOR. β-actin was used as a control for quantification. C: PBS; SJWE: Ethanol extract of St. John’s Wort 50 mg/kg. Significantly different from control by Student’s *t*-test (*, *p* < 0.05; **, *p* < 0.01; ***, *p* < 0.001; ****; *p* < 0.0001).

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
