# Peer review of "St. John’s Wort Suppresses Growth in Triple-Negative Breast Cancer Cell Line MDA-MB-231 by Inducing Prodeath Autophagy and Apoptosis"

_nutrients, 2020, doi:10.3390/nu12103175_

Round 1
Reviewer 1 Report
In this manuscript titled “St. John’s Wort suppresses growth in triple-negative breast cancer cell line MDA-MB-231 by inducing pro-3 death autophagy and apoptosis”, You et.al have investigated the role of St. John’s Wort ethanol extract (SJWE) in regulating the proliferation of triple negative breast cancer cell line MDA-MB-231. First, they found SJWE inhibited the growth of the cell line in a dose dependent manner which was coupled with increased autophagy and apoptosis level. By means of chemical inhibition on autophagy, the authors have proved that apoptosis and growth inhibition was mediated by the pro-autophagy effect of SJWE. In the end they confirmed the anti-tumor effect of SJWE in vivo.
The study provided an interesting point of view on how SJWE regulates cancer cell survival by affecting AMPK-mTOR signaling and related survival pathways. However, there are still some points needed to be elucidated:
- In this manuscript, the authors proposed that SJWE might be a potential anticancer therapy for TNBC. However, to make this statement valid, the effect on cell proliferation, autophagy, or apoptosis of SJWE with normal cell lines such as MCF-10A should be performed to confirm the anti-tumor preference of SJWE.
- What is the difference between the cell viability of Figure 1A and the proliferation of Figure 1B? Did the authors perform assays such as CellTiter-Glo to measure the viability of cells? If 1A is also cell counting, please keep the consistency when referring to the same method; if not, please specify the reason to use a different method.
- In the apoptosis assay of Figure 3A, after 24 hours of SJWE treatment 86% of cells were in apoptosis in the 50ug/ml group, showing a very strong apoptosis inducing effect. However, it is a bit strange that the cell viability didn’t change at all under the same condition in Figure 1A, and still the cell number was 20% of control after 5 days. Could the authors give an explanation on this finding, and provide some additional evidence of apoptosis such as Caspase activation or PARP cleavage at the time point of 24h?
- The authors claimed that SJWE increased autophagy by inhibiting PI3K-mTOR signaling. In Figure 3C, p-mTOR level did decreased with SJWE, however the p-PI3K/PI3K level seems increased as weaker PI3K and unchanged p-PI3K in 50ug/ml sample. I don’t really understand how the authors have calculated a lower level p-PI3K/PI3K in the quantification. Same in Figure 4D, the PI3K level of SJWE without 3-MA seems lower than control, and the authors still got an increased p-PI3K/PI3K in the quantification.
- In Figure 4, the authors used 3-MA, a Vps34 and PI3Kγ inhibitor to suppress autophagy. When studying the cell death induced by autophagy, it is recommended to perform also genetic inhibition of a few key components of the pathways to validate the link. In this study, it would be nice if the authors could use shRNA/siRNA/CRISPR-KO to block the autophagy pathway and check the cell proliferation, autophagy, and apoptosis when cells treated with SJWE. Further, if the authors want to make the statement in Line 175-176 that SJWE regulates PI3K/mTOR pathway by affecting autophagy, similar experiment should also be done by checking p-PI3K/PI3K and p-mTOR/mTOR with genetic blocking of autophagy when treated with SJWE.
Minor points:
- Line 68, MDA-MB 231 needs to be corrected.
- Line 128-129, “In addition, Atg5 protein level…than in the control”, this sentence need to be rephrased.
- Line 166, “we treated with SJWE…” MDA-MB-231 cells were treated?
- Line 171, “…the activation of PI3K (Figure 3B)…” should be Figure 3C.
- Line 237, “find our” should be “find out”.
- Line 257, “mTOR/mTOR” should be “p-mTOR/mTOR”.
Author Response
Please check the attached file. Thank you.
1. In this manuscript, the authors proposed that SJWE might be a potential anticancer therapy for TNBC. However, to make this statement valid, the effect on cell proliferation, autophagy, or apoptosis of SJWE with normal cell lines such as MCF-10A should be performed to confirm the anti-tumor preference of SJWE.
: We appreciate and were impressed your meticulous advice. We are afraid that we have not done this study with MCF-10A. Unfortunately, considering the timeline of revision, we were unable to confirm the anti-tumor preference of SJWE with MCF-10A. However, we will remember this comment to try the next time for quality research.
In supporting the reviewer’s opinion, we tested the proliferation assay using the human keratinocytes, HaCaT cells. The proliferation in the HaCaT cells wasn’t altered by SJWE treatment. We assumed that SJWE might affect cancer cell proliferation.
2. What is the difference between the cell viability of Figure 1A and the proliferation of Figure 1B? Did the authors perform assays such as CellTiter-Glo to measure the viability of cells? If 1A is also cell counting, please keep the consistency when referring to the same method; if not, please specify the reason to use a different method.
: To identify the effect of SJWE on cell health and growth we tested cell viability and proliferation. For cell viability, cells were treated with SJWE for 24h after cells were 80% confluent and then stained living cells in a population using MTT assay (Figure 1A). To measure the effect of SJWE on cell division, cells were treated with SJWE for 5 days and then the living cells were counted using hemocytometer. In the cell proliferation assay, we double checked the cell toxicity of SJWE on the 5th day of treatment, which shows the cell count was less than that seeded at the beginning. In the revised manuscript, we updated the cell viability and proliferation methods in greater detail (Line 69-70).
3. In the apoptosis assay of Figure 3A, after 24 hours of SJWE treatment 86% of cells were in apoptosis in the 50ug/ml group, showing a very strong apoptosis inducing effect. However, it is a bit strange that the cell viability didn’t change at all under the same condition in Figure 1A, and still the cell number was 20% of control after 5 days. Could the authors give an explanation on this finding, and provide some additional evidence of apoptosis such as Caspase activation or PARP cleavage at the time point of 24h?
: We agree with the reviewer. We treated SJWE in the TNBC cells for 24h and then tested cell viability and apoptosis using MTT and Annexin V assay, respectively. The MTT assay assesses cell metabolic activity and Annexin-V assay measures the loss of phospholipid asymmetry in the cell membrane (Analyst, 2016:14;414(23):6408-6415). That’s why the measurements show different views and different results.
Another explanation for the differences is that we can choose four quadrants by ourselves. After measuring apoptosis with Muse® Annexin V & Dead cell, we can choose a line to make four quadrants. That’s why every research team showed different quadrants (Int J Mol Sci, 2019:26;20(3):521, Int J Mol Med, 2014:34;(5):1249-56, BMC Complement Altern Med, 2017:28;17(1):236). If we choose an equal quadrant or diagonal line instead of the current line, we will get different results that have an earlier apoptotic cell population.
Based on the reviewer’s opinion, we provided cleaved caspase 3 results. SJEW showed an increase in the expression of cleaved caspase 3 (see below), which shows the same pattern of increase of the apoptotic cell population (Figure 3A).
4. The authors claimed that SJWE increased autophagy by inhibiting PI3K-mTOR signaling. In Figure 3C, p-mTOR level did decreased with SJWE, however the p-PI3K/PI3K level seems increased as weaker PI3K and unchanged p-PI3K in 50ug/ml sample. I don’t really understand how the authors have calculated a lower level p-PI3K/PI3K in the quantification. Same in Figure 4D, the PI3K level of SJWE without 3-MA seems lower than control, and the authors still got an increased p-PI3K/PI3K in the quantification.
: We regret that we made a mistake by mixing the presentation of p-PI3K and PI3K images in the Figure 3C and 4D. Actually, we observed the decrease of p-PI3K and p-mTOR expression in SJWE treated groups. We changed the image of the western band and showed each of p-form and t-form expressions, respectively in a bar graph. Moreover, we showed raw data of all images of p-PI3K and PI3K expression to be certain of our results.
5. In Figure 4, the authors used 3-MA, a Vps34 and PI3Kγ inhibitor to suppress autophagy. When studying the cell death induced by autophagy, it is recommended to perform also genetic inhibition of a few key components of the pathways to validate the link. In this study, it would be nice if the authors could use shRNA/siRNA/CRISPR-KO to block the autophagy pathway and check the cell proliferation, autophagy, and apoptosis when cells treated with SJWE. Further, if the authors want to make the statement in Line 175-176 that SJWE regulates PI3K/mTOR pathway by affecting autophagy, similar experiment should also be done by checking p-PI3K/PI3K and p-mTOR/mTOR with genetic blocking of autophagy when treated with SJWE.
:We agree with the reviewer’s opinion and recommendation. We are aware that genetic inhibition like siRNA is a good tool to check the molecular pathway study. In this study, we focused on the effect of SJWE on the autophagy-induced apoptosis. Our next research plan is showing that SJWE is therapeutic strategy against TNBC cells via targeting the PI3K/mTOR pathway. To test this, we will use PI3K/mTOR dual inhibitor, LY3023414 and siRNA tools. In supporting the reviewer’s opinion, we removed “which regulates the PI3K/mTOR pathway” in the revised manuscript (Line 179).
Minor points:
1. Line 68, MDA-MB 231 needs to be corrected.
→ Typo fixed to “MDA-MB-231”. Thanks.
2. Line 128-129, “In addition, Atg5 protein level…than in the control”, this sentence need to be rephrased.
→ We changed the sentence “In addition, autophagy related 5 (Atg5) was increased, whereas Atg12 was reduced by SJWE treatment.”. Thanks.
3. Line 166, “we treated with SJWE…” MDA-MB-231 cells were treated?
→ We changed the sentence “MDA-MB-231 cells were treated”. Thanks.
4. Line 171, “…the activation of PI3K (Figure 3B),” should be Figure 3C.
→ It fixed to “Figure 3C”. Thanks.
5. Line 237, “find our” should be “find out”.
→ It fixed to “find out”. Thanks.
6. Line 257, “mTOR/mTOR” should be “p-mTOR/mTOR”.
→ It fixed to “p-mTOR/mTOR”. Thanks.

Reviewer 2 Report
In this manuscript the authors describe the effect of St. John Wort on cell proliferation of triple negative breast cancer cells. They show that SJWE reduce the cell proliferation by inducing pro-death autophagy and apoptosis.
In Fig.2A, upper panel I’m not able to appreciate LC3-I and II altered levels and the same into the western blot in fig 3B (excluding p-Bad).
The manuscript is clear and well written but the results, including the western blot, not always show that the authors describe into the text and must be ameliorated. Even if the study is poorly examined in depth, it shows the effect of SJWE on triple negative breast cancer cells and it is suitable to be published on this journal after minor revisions.
Author Response
Please see the upload word file.
1. In Fig.2A, upper panel I’m not able to appreciate LC3-I and II altered levels and the same into the western blot in fig 3B (excluding p-Bad).
: Thank you for your comments. In support of the reviewer’s opinion, we changed LC3 band in the Figure 2A and Bcl-xL, Bcl-2, Bad, and Bax in the Figure 3B. In addition, we provided raw data of all images of Bcl-xL, Bcl-2, Bad, and Bax.

Reviewer 3 Report
In the manuscript titled, “St. John’s Wort suppresses growth in triple-negative breast cancer cell line MDA-MB-231 by inducing pro-death autophagy and apoptosis” the authors demonstrated that St. John’s Wort extract (SJWE) has inhibitory effect on triple negative breast cancer cells through activation of autophagy and apoptosis, and inhibition of proliferation in both in-vitro and in-vivo studies. The authors should correlate the different data they included in the manuscript. Overall, the manuscript requires major revision and my comments are below.
- Difference in the levels of PI3k and p-PI3k expression status between in-vitro and in-vivo studies should be addressed. The individual level of these proteins should to be considered together with PI3K/p-PI3K ratio.
- In the in-vivo xenograft study, individual levels of PI3k and p-PI3k are increased with SJWE treatment compared to control which contradict with the hypothesis the authors presented in the manuscript. The authors should provide a reasonable explanation for these findings or repeat this part of the experiment.
- In the in-vitro experiment, p-PI3k levels look more or less same between treatment and control even though there is a minimal decrease in total PI3K level. The authors should provide a reasonable explanation for these findings.
- The authors should provide the particular subunit of the PI3K used in the study. Looking at the antibody details provided in the supplemental section; the subunit is p85. The authors should update this or add molecular weight of each protein in the immunoblot pictures.
- mTOR expression status in in-vivo studies is increased but decreased in in-vitro studies. The authors should provide a reasonable explanation for this difference.
- In figure 2A and 2B, actin immunoblot pictures are similar. If these experiments were performed at the same time, then these sub-figures should be combined in a single sub-figure. Same applies for figure 3B and 3C. Figure legends also should be updated.
- TUNEL assay pictures in figure 5C should be enlarged to show the DAB staining of apoptotic cells and quantify as percentage of apoptotic cells.
- Immunoblot pictures in figure 5E and 5F should have similar brightness and clear blot images.
Author Response
Please see the upload word file.
1. Difference in the levels of PI3k and p-PI3k expression status between in-vitro and in-vivo studies should be addressed. The individual level of these proteins should to be considered together with PI3K/p-PI3K ratio.
-In the in-vivo xenograft study, individual levels of PI3k and p-PI3k are increased with SJWE treatment compared to control which contradict with the hypothesis the authors presented in the manuscript. The authors should provide a reasonable explanation for these findings or repeat this part of the experiment.
: We tried many times, but both p-PI3K and PI3K expressions in SJWE treated tumor tissue was higher than that in control tumor tissue. In many studies, the p-PI3K/PI3K ratio is considered more important than individual images. The p-PI3K/PI3K ration decreased in SJWE-treated tumor tissue in vivo. We added the each of the p-form and t-form of PI3K expression in a bar graph and showed more images of p-PI3K and PI3K.
- In the in-vitro experiment, p-PI3k levels look more or less same between treatment and control even though there is a minimal decrease in total PI3K level. The authors should provide a reasonable explanation for these findings.
: We regret that we made a mistake by mixing the presentation of p-PI3K and PI3K images in the Figure 3C and 4D. Actually, we observed the decrease of p-PI3K and p-mTOR expression in SJWE treated groups. We changed the image of the western band and showed each of p-form and t-form expressions, respectively in a bar graph. Moreover, we showed raw data of all images of p-PI3K and PI3K expression to be certain of our results.
- The authors should provide the particular subunit of the PI3K used in the study. Looking at the antibody details provided in the supplemental section; the subunit is p85. The authors should update this or add molecular weight of each protein in the immunoblot pictures.
: Thank you for catching our mistakes. It was added in the supplemental information (Table S1) and methods part and figure legends in the revised manuscript (Line 81, 155, 184, and 209-210).
2. mTOR expression status in in-vivo studies is increased but decreased in in-vitro studies. The authors should provide a reasonable explanation for this difference.
: In supporting the reviewer’s opinion, we added a bar graph of p-mTOR and mTOR in the Figure 3C, 4D, and 5F in the revised manuscript (Line 179).
3. In figure 2A and 2B, actin immunoblot pictures are similar. If these experiments were performed at the same time, then these sub-figures should be combined in a single sub-figure. Same applies for figure 3B and 3C. Figure legends also should be updated.
: In response to the reviewer’s suggestion, we changed the β-actin immunoblot picture in Figure 2B and 3B. We also updated figure legends (Line 153).
4. TUNEL assay pictures in figure 5C should be enlarged to show the DAB staining of apoptotic cells and quantify as percentage of apoptotic cells.
: Thank you for your suggestion. In response to the reviewer’s suggestions, we enlarged the images and quantified the relative TUNEL positive cells in the Figure 5C.
5. Immunoblot pictures in figure 5E and 5F should have similar brightness and clear blot images.
: In response to the reviewer’s suggestion, we modified the brightness and contrast in the Figure 5E and 5F.

Round 2
Reviewer 1 Report
The authors have explained all questions well with additional data that incorporated, I recommend to accept the revised manuscript.